# A Quantitative ELISA Protocol for Detection of Specific Human IgG against the SARS-CoV-2 Spike Protein

**DOI:** 10.3390/vaccines9070770

**Published:** 2021-07-09

**Authors:** Rémi Vernet, Emily Charrier, Julien Grogg, Nicolas Mach

**Affiliations:** 1Department of Oncology, Geneva University Hospitals and Medical School, 1211 Geneva, Switzerland; emily.charrier@unige.ch (E.C.); nicolas.mach@hcuge.ch (N.M.); 2Center for Translational Research in Onco-Hematology, Division of Oncology, Geneva University Hospital and University of Geneva, 1211 Geneva, Switzerland; 3MaxiVAX SA, 1202 Geneva, Switzerland; jgrogg@maxivax.ch

**Keywords:** SARS-CoV-2, COVID-19, ELISA, antibodies

## Abstract

Severe acute respiratory syndrome coronavirus 2 (SARS-CoV-2) has caused a worldwide pandemic with at least 3.8 million deaths to date. For that reason, finding an efficient vaccine for this virus quickly became a global priority. The majority of vaccines now marketed are based on the SARS-CoV-2 spike protein that has been described as the keystone for optimal immunization. In order to monitor SARS-CoV-2 spike-specific humoral responses generated by immunization or infection, we have developed a robust and reproducible enzyme-linked immunosorbent assay (ELISA) protocol. This protocol describes a method for quantitative detection of IgG antibodies against the SARS-CoV-2 spike protein using antigen-coated microtiter plates. Results showed that antibodies could be quantified between the range of 1.953 ng/mL to 500 ng/mL with limited inter- and intra-assay variability.

## 1. Introduction

COVID-19, the disease caused by severe acute respiratory syndrome coronavirus 2 (SARS-CoV-2), has triggered a worldwide pandemic with over 176 million confirmed cases globally and more than 3.8 million deaths as of June 2021 [1]. In response to this major public health crisis, global efforts have been dedicated to developing different SARS-CoV-2 vaccines using a variety of technologies [2,3,4,5,6] and the SARS-CoV-2 spike protein (S protein). Indeed, the SARS-CoV-2 spike protein has often been described as the keystone for obtaining an optimal and efficient immunization by eliciting neutralizing antibodies and targeting the structural domains of the S protein: S1 RBD, S1 N-terminal domain, or the S2 region [7,8].

Following vaccination or infection, cellular and humoral responses are generated by the host immune system [9,10]. Immediately, innate immunity defenses are triggered to slow down or inhibit initial infection by protecting cells from infection or by eliminating virus-infected cells. This initial reactivity allows time for the adaptive immune response to begin. The adaptive immune response is based on T cell- and B cell-mediated responses. Firstly, T cell-mediated response will take place, principally by recognition and destruction of virus-infected cells [11]; secondly, specific antiviral antibodies released by mature secreting plasma B cells and commonly named immunoglobulins (Ig) create an immune-protective barrier against infection [12].

Consequently, to obtain a durable protective immunity, memory B cells and/or memory T cells must be obtained [13]. Memory cells are lasting immune cells capable of recognizing foreign proteins to which they have previously been exposed. These immune cells will facilitate a faster secondary response when the antigen is encountered on a subsequent occasion. Memory plasma cells go on to continuously secrete antibodies which allow the immune system to maintain a stable humoral immunological memory over long periods. In this way, multiple studies are investigating seroprevalence within different populations, for example, in patients with cancer [14] and within different environmental contexts [15,16]. These studies are mainly focused on IgG [17] and IgM patterns for each cohort or in relation to COVID-19 symptoms and severity [18,19]. Given the development of many different vaccines, they are also evaluating their effectiveness to generate short- and long-term immune responses. 

For this research, the scientific community requires sound operational diagnostic tools [20]. Nucleic acid amplification, such as polymerase chain reaction (PCR), can be used for the detection of the viral genome, but several time-consuming steps need to be performed. This test is optimal and is generally recommended for the diagnosis of an acute SARS-CoV-2 infection. However, specific antibody-based technology will allow detection of specific antibodies directed against the SARS-CoV-2 virus, and this serological testing may be considered more reliable for the diagnosis of suspected patients presenting with negative viral genomic results and for the analysis of asymptomatic infections [21]. Specific antibody quantification technology will therefore prove essential in studying the effectiveness of the various vaccinations commercialized or under development. To date, there is a lack of commercially available products in different countries; most commercialized kits were only available outside the European area. Moreover, no detailed protocols have been yet shared by the scientific community.

In the present study, we have developed a robust and reproducible enzyme-linked immunosorbent assay (ELISA) protocol. This protocol describes a method for quantitative detection of IgG antibodies against the SARS-CoV-2 spike protein with limited inter- and intra-assay variability. This limited variability has been validated with batches of independently prepared standard curves.

## 2. Materials and Methods

### 2.1. SARS-CoV-2 Spike Protein-Specific IgG Quantification by ELISA

The following steps were carried out in compliance with good laboratory practices (GLP):Clear Flat-Bottom Immuno Nonsterile MEDISORP 96-Well Plates (Thermo Fisher Scientific, Waltham, MA, USA, 467320) were coated with 100 nanograms (ng) of SARS-CoV-2 spike protein S1/S2 (S-ECD) (aa11-1208) (Thermo Fisher Scientific, RP87680) in 100 microliters (μL) phosphate-buffered saline (PBS 1X) (Sigma Aldrich, St. Louis, MO, USA, D8537–500 ML) per well for 16 to 18 h at 4 °C.1X wash buffer was prepared from 25X concentrated solution (Thermo Fisher Scientific, 3501004) by allowing it to reach room temperature (RT), then mixing to ensure that any precipitated salts had been dissolved before diluting with deionized water (Reactolab, Chroma, Servion, Switzerland, 99389.9010). Plates were washed 4 times for 5 min with 250 μL of the 1X wash buffer under 200 rpm agitation using a plate orbital shaker. Plates were blocked with 250 μL of PBS 1X + 0,05% Tween 20 (AppliChem, Darmstadt, Germany, A1389,0500) + 1% Bovine Serum Albumin (BSA) (Sigma Aldrich, A4503-50G) for 1 h at RT under 200 rpm agitation using a plate orbital shaker. Plates were incubated either with 100 μL of samples or, for the standard curve, 100 μL of primary antibody S-RBD Chimeric Recombinant Mouse Monoclonal Antibody dilution (D005) (Thermo Fisher Scientific, MA5-35958). The standard curve was prepared using serial dilution from 500 ng/mL until 1.953 ng/mL. The samples and antibody were diluted in blocking buffer to obtain 100 μL of each analyte. These were then incubated for 90 min at RT under 200 rpm agitation using a plate orbital shaker. Plates were washed 4 times for 5 min with 300 μL of 1X wash buffer under 200 rpm agitation using a plate orbital shaker.Plates were incubated with 100 μL of a 1 μg/mL secondary antibody, Mouse anti-Human IgG1 Fc secondary antibody, HRP solution (Thermo Fisher Scientific, A-10648) for the detection of bound antibodies for 1 h at RT under 200 rpm agitation using a plate orbital shaker.Plates were washed 4 times for 5 min with 300 μL of 1X wash buffer under 200 rpm agitation using a plate orbital shaker.Plates were revealed by chromogenic revelation using 100 μL of 1X TMB substrate (Thermo Fisher Scientific, SB01) for 30 min at RT in the dark and stopped with 100 μL of 1X stop solution (Thermo Fisher Scientific, SS01). Plates were read immediately with a SpectraMax reader for optical density/absorbance of the samples at 450 nm. The concentration of the samples was then calculated using the equation of the standard curve and taking into account the dilution of each sample.

### 2.2. Data Analysis

Data analyses were performed using SoftMax Pro 7.1 (Molecular Devices Software) and Microsoft^®^ Excel^®^ 2013 (Microsoft Office Professional Plus 2013).

## 3. Results

### 3.1. Reproducibility of SARS-CoV-2 Spike Protein-Specific IgG Quantification by ELISA

This protocol has been designed on the hypothesis that a quantitative, sensitive, and reproducible homemade ELISA is feasible with individual commercially available consumables.

This standardized protocol allows quantitative detection of anti-SARS-CoV-2 IgG antibodies in serum samples. Data presented in Table 1 shows optical density (O.D.) values for nine standard concentration curves across the range of 1.953 ng/mL to 500 ng/mL human-specific IgG against the SARS-CoV-2 spike protein. Four independent standard curves were developed and run over two independent experiments on two different days and on different plates. They are all represented by a 5 Parameter Logistic curve in Figure 1.

Reproducibility within the assay was assessed over these four independent nine-standard concentration curves. Each standard concentration curve was performed in triplicate. Results obtained show an overall calculated intra-assay coefficient of variation (CV) of 1.53% (range of 1.0% to 1.8%).

Reproducibility of the standard curves between assay to assay, performed on the same day and on two distinct plates, gave rise to a CV of 2.8% for the first two plates (range of 0.3% to 5.1%) and 2.9% for the second two plates (range of 0.3% to 7.3%). 

All of these intra-assay O.D. values, whether inside the same plate or during the same run, show robust reproducibility for all of the nine-standard concentration curves.

Inter-assay precision within our laboratory was calculated between all of the values obtained during these four independent experiments and gave rise to an overall interassay CV of 23.4% (range of 0.3% to 39.3%). This overall inter-assay CV shows a proportionally inverse increase compared to the nine-standard concentration curves, meaning that inter-assay precision may be around 30% when concentrations lower than 15 ng/mL are analyzed on different days.

### 3.2. Analytical Sensitivity of SARS-CoV-2 Spike Protein-Specific IgG Quantification by ELISA

Regarding the sensitivity of this assay, twelve values were obtained using the dilution buffer (Table 2). These values were obtained at the same time and in the same independent manner as the previously generated results used for the validation of this assay reproducibility.

The lower limit of detection (LLOD) was defined as the overall mean O.D. value plus ten standard deviations (SD) as obtained with the dilution buffer. The LLOD for this assay was validated with an O.D. of approximately 0.004 at 450 nm and is therefore considerably lower than the lowest O.D. tested concentration of this assay (1.953 ng/mL). Effectively, the mean overall O.D. for the 1.953 ng/mL is 0.1 (SD 0.039). By extrapolating, we can say that the LLOD is, therefore, below the 1.953 ng/mL concentration.

In parallel, negative human serum control was used in order to check whether the background is generated with this analyte. The result obtained was similar to the LLOD, with an O.D. of 0.004 at 450 nm.

## 4. Discussion

This standardized quantitative enzyme-linked immunosorbent assay protocol for detection of specific human IgG against the SARS-CoV-2 spike protein is an assay designed to detect and quantify the level of Human IgG anti-SARS-CoV-2 spike protein. 

All of the components (plastic plates, coating buffer, concentration of the coated protein/secondary antibody) and parameters within each step of this protocol have been optimized to allow researchers to perform a quantitative, sensitive, and reproducible homemade ELISA. Parameters have been selected based on the broad range of obtained O.D.s for the standard curve and on the lowest background generated by plastic and chemical consumables. Moreover, different volumes of samples have been tested in order to minimize the use of the rare and precious samples that can be obtained from studied patients and/or laboratory animals. Indeed, a maximum of 5 μL is needed for the less concentrated samples.

The results obtained leave us confident about the good reproducibility of this quantitative ELISA protocol. However, inter-assay precision within our laboratory shows overall inter-assay CV higher than 30% when analyzed concentrations are smaller than 15 ng/mL. While notably, these data fall within the range of commercialized products but with a much lower cost and with worldwide availability of products. 

The limitation of the ELISA assay technology is that the scientific community should keep in mind that when comparing same-subject samples, it is preferable to analyze them at the same time. Furthermore, for long-term studies, it may be advisable to include samples with previously determined concentrations as internal positive quality controls of assay performance. These controls should be used to determine if a run is acceptable or needs to be redone.

Finally, this protocol presents the potential to adapt the revelation antibodies to the species studied, allowing it to be applied to many different sample species. This was demonstrated by our previous study [5], where we analyzed mice samples. We obtained qualitative data from each mice sample and were able to compare baseline samples with sacrifice samples. It can also be useful for seroconversion studies for which IgG and IgM levels need to be quantified separately. In this way, identically prepared and diluted samples can be analyzed and compared at the same time and on the same coated plate, using several secondary revelation antibodies.

Within the pandemic scenario, we believe that a balance must be achieved between rapid development and obtaining robust and reproducible analytical methods. Indeed, it is essential for the scientific community to have resilient operational tools in order to generate comparable data and minimize variability. In the context of the COVID-19 pandemic, this quantitative ELISA protocol can be utilized for research used only to meet this demand without compromising good laboratory practices.

## 5. Conclusions

In conclusion, this protocol describes a method for quantitative detection of IgG antibodies against the SARS CoV 2 spike protein using antigen-coated microtiter plates. Results showed that antibodies could be quantified between the range of 1.953 ng/mL to 500 ng/mL with limited inter- and intra-assay variability.

## Figures and Tables

**Figure 1 vaccines-09-00770-f001:**
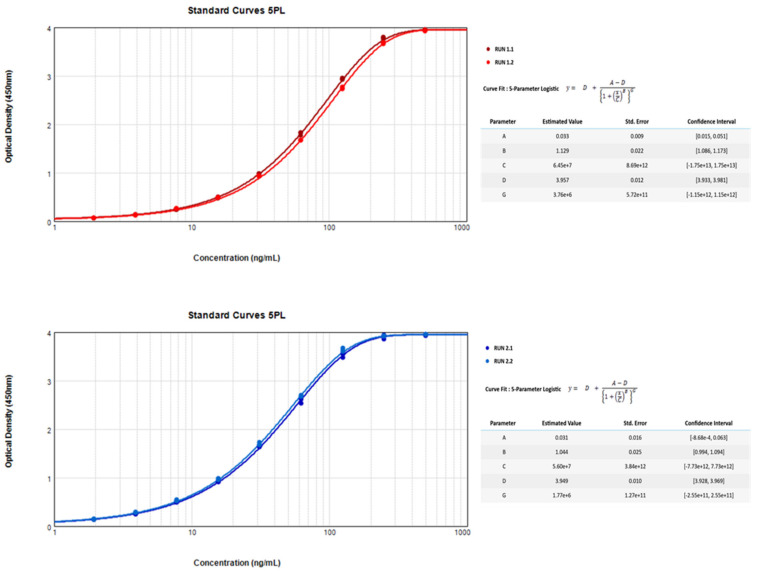
Graphical representation by a 5 Parameter Logistic (5PL) Curve of the Results for the various standard curves across the range of 1.953 ng/mL to 500 ng/mL human-specific IgG against the SARS-CoV-2 spike protein.

**Table 1 vaccines-09-00770-t001:** Results for the various standard curves over the range of 1.953 ng/mL to 500 ng/mL.

**Concentration** **(ng/mL)**	**Run 1.1**						**Run 1.2**						**Intra Run 1 Value**		
	**O.D. 1–450 nm**	**O.D. 2–450 nm**	**O.D. 3–450 nm**	**Mean O.D. Value**	**SD**	**CV**	**O.D. 1–450 nm**	**O.D. 2–450 nm**	**O.D. 3–450 nm**	**Mean O.D. Value**	**SD**	**CV**	**Mean O.D. Value**	**SD**	**CV**
**500**	3.933	3.953	3.953	3.946	0.012	0.3	3.924	3.935	3.924	3.927	0.006	0.2	3.937	0.013	0.3
**250**	3.748	3.795	3.775	3.773	0.024	0.6	3.671	3.648	3.680	3.666	0.016	0.4	3.719	0.061	1.6
**125**	2.916	2.925	2.947	2.929	0.016	0.6	2.764	2.732	2.734	2.743	0.018	0.7	2.836	0.103	3.6
**62.5**	1.772	1.823	1.828	1.808	0.031	1.7	1.675	1.673	1.673	1.674	0.001	0.1	1.741	0.076	4.4
**31.25**	0.942	0.975	0.984	0.967	0.022	2.2	0.931	0.918	0.923	0.924	0.007	0.7	0.945	0.028	2.9
**15.625**	0.478	0.488	0.497	0.488	0.009	1.9	0.488	0.483	0.490	0.487	0.004	0.7	0.487	0.006	1.3
**7.813**	0.234	0.242	0.244	0.240	0.005	2.3	0.260	0.265	0.255	0.260	0.005	1.8	0.250	0.012	4.8
**3.906**	0.134	0.138	0.131	0.135	0.004	2.7	0.126	0.125	0.132	0.128	0.004	3.0	0.131	0.005	3.9
**1.953**	0.066	0.067	0.063	0.065	0.002	3.0	0.060	0.060	0.061	0.060	0.001	1.4	0.063	0.003	5.1
					**Mean CV = 1.7**							**Mean CV = 1.0**			**Mean CV = 2.8**
**Concentration** **(ng/mL)**	**Run 2.1**						**Run 2.2**						**Intra Run 2 Value**		
	**O.D. 1–450 nm**	**O.D. 2–450 nm**	**O.D. 3–450 nm**	**Mean O.D. Value**	**SD**	**CV**	**O.D. 1–450 nm**	**O.D. 2–450 nm**	**O.D. 3–450 nm**	**Mean O.D. Value**	**SD**	**CV**	**Mean O.D. Value**	**SD**	**CV**
**500**	3.945	3.922	3.941	3.936	0.012	0.3	3.952	3.949	3.952	3.951	0.002	0.0	3.944	0.011	0.3
**250**	3.847	3.932	3.916	3.898	0.045	1.1	3.918	3.905	3.927	3.917	0.011	0.3	3.908	0.031	0.8
**125**	3.474	3.538	3.573	3.528	0.050	1.4	3.592	3.632	3.672	3.632	0.040	1.1	3.580	0.070	1.9
**62.5**	2.530	2.619	2.657	2.602	0.065	2.5	2.701	2.670	2.679	2.683	0.016	0.6	2.643	0.061	2.3
**31.25**	1.626	1.644	1.658	1.643	0.016	1.0	1.728	1.672	1.677	1.692	0.031	1.9	1.667	0.035	2.1
**15.625**	0.908	0.909	0.932	0.917	0.013	1.5	0.980	0.948	0.979	0.969	0.018	1.9	0.943	0.032	3.4
**7.813**	0.482	0.506	0.497	0.495	0.012	2.5	0.524	0.519	0.539	0.527	0.010	2.0	0.511	0.020	4.0
**3.906**	0.244	0.244	0.246	0.244	0.001	0.4	0.285	0.274	0.278	0.279	0.006	2.1	0.262	0.019	7.3
**1.953**	0.141	0.140	0.128	0.136	0.007	5.2	0.145	0.139	0.133	0.139	0.006	4.2	0.138	0.006	4.4
					**Mean CV = 1.8**							**Mean CV = 1.6**			**Mean CV = 2.9**
**Concentration** **(ng/mL)**					**Inter-Assay to Assay Value**										
					**Mean O.D. Value**		**SD**					**CV**			
500					3.940		0.012					0.3			
250					3.813		0.108					2.8			
125					3.208		0.397					12.4			
62.5					2.192		0.476					21.7			
31.25					1.306		0.378					29.0			
15.625					0.715		0.239					33.4			
7.813					0.380		0.137					36.1			
3.906					0.196		0.069					35.4			
1.953					0.100		0.039					39.3			
					**Mean CV = 23.4**										

**Table 2 vaccines-09-00770-t002:** Evaluation of the Lower Limit of Detection.

			Concentration 0 ng/mL			
	O.D. 1–450 nm	O.D. 2–450 nm	O.D. 3–450 nm	Mean O.D. Value	SD	LLOD
RUN 1.1	−0.000199	0.000600	0.000400			
RUN 1.2	0.000600	0.000200	0.000400	0.00016	0.00039	0.0040
RUN 2.1	−0.000433	0.000007	0.000367			

RUN 2.2	0.000433	−0.000567	0.000133			

## Data Availability

Data can be accessed upon request.

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
