# Peer review of "A Quantitative ELISA Protocol for Detection of Specific Human IgG against the SARS-CoV-2 Spike Protein"

_vaccines, 2021, doi:10.3390/vaccines9070770_

Round 1

Reviewer 1 Report

Vernet et al present an ELISA method to perform specific quantification of IgG  to SARS-CoV-2 spike protein. Although the method presented is not novel they show good CV across many runs. The methods are detailed and can be easily adapted by any researchers which is commendable. The biggest drawback of this study, the authors do not test their method on clinical serum samples but only use positive and negative controls. In addition please find some of my concerns

  • Introduction perhaps can describe previous efforts in this space? perhaps also explain the knowledge gaps and why there is need for our study?
  • I would like to see some real world serum samples tested, A gold standard sample from https://www.nibsc.org/science_and_research/idd/cfar/covid-19_reagents.aspx could be used to validate the results. Human serum is far more heterogenous than a targeted monoclonal antibody 
  • The discussion needs to address your method in relation to other previously described similar methods,  a paragraph on limitations of your method could also benefit the discussion.

Author Response

Dear Reviewer,

Thank you for your pertinent comments. Please find below our answers:

Vernet et al present an ELISA method to perform specific quantification of IgG to SARS-CoV-2 spike protein. Although the method presented is not novel they show good CV across many runs. The methods are detailed and can be easily adapted by any researchers which is commendable. The biggest drawback of this study, the authors do not test their method on clinical serum samples but only use positive and negative controls. In addition please find some of my concerns

Introduction perhaps can describe previous efforts in this space? perhaps also explain the knowledge gaps and why there is need for our study?

We agreed, and the introduction has been improved with the following text:

“To date, there is a lack in the commercially available products in different countries; most commercialized kits were only available outside Europe area. Moreover, no detailed protocols have been yet shared by the scientific community.”

These points/explanations have also been addressed during the discussion:

“While notably, these data fall within the range of commercialized products but with a much lower cost and with worldwide availability of products.”

“this protocol presents the potential to adapt the revelation antibodies to the species studied, allowing it to be applied to many different sample species”

Finally, we want to allow the research community to have the possibility to study different species samples with the same protocol (excepted for the secondary antibody).

I would like to see some real world serum samples tested, A gold standard sample from https://www.nibsc.org/science_and_research/idd/cfar/covid-19_reagents.aspx could be used to validate the results. Human serum is far more heterogenous than a targeted monoclonal antibody

We agree that human serum samples are far more heterogenous than a targeted monoclonal antibody but all the ELISA kit are built in this way.

As this manuscript is a Protocol, and not an Article, we have decided to show only negative values of human serum samples for the LLOD in order to describe parameters of this assay.

For experimental data, we used this protocol in a previous study as mentioned in the discussion:

“Finally, this protocol presents the potential to adapt the revelation antibodies to the species

studied, allowing it to be applied to many different sample species. This was demonstrated by our previous study”

Lastly, this protocol will be used for upcoming analysis. In all studies, as requested in the discussion a control must be used. “Furthermore, for long term studies it may be advisable to include samples with previously determined concentrations as internal positive quality controls of assay performance. These controls should be used to determine if a run is acceptable or needs to be redone.”

The discussion needs to address your method in relation to other previously described similar methods, a paragraph on limitations of your method could also benefit the discussion.

As there were no previously published similar methods it is difficult to do comparison.

Therefore, we wrote “these data fall within the range of commercialized products”.

For the limitation of this assay a paragraph has been created as follow: “Limitation of the ELISA assay technology is that the scientific community should keep in mind ….”.

Kind Regards,

Rémi Vernet

Reviewer 2 Report

Authors present a protocol for detection of IgG ab with Elisa for SARS-CoV-2 spike protein

Major Comments:

Introduction is unnecessarily long without any particular focus on the protocol or product development. There is no need for the basics of virology, this could be an interesting paper for the advances learners and should be kept so.

2nd and 3rd paragraphs of introduction can be replaced with the protocol specific introduction or deleted all together.

Results:

Prior to the presentation of the results, authors should present a hypothesis.

If this is a feasibility study, then authors should mention so (even for the protocol development)

Why did the authors use coefficient of variation to assess the reproducibility?

Standard error in percent or interclass correlation coefficient could be used for the measurement of reproducibility (considered superior)

I would strongly suggest that the authors re-run the analysis with above mentioned statistical methods

I agree with the authors that we need a balance between rapid development and robust reproducible research, but the first dictum of medicine (Do no harm) should be followed. Unless, the results are plausible and supported by legit statistical methods it would not be prudent to over-confident with the clinical utility of the assay.

In the introduction, the authors should also provide some background about the development of this protocol.

Under discussion the authors need to compare the results with other assays that are available or under development at this time. If there are none, then the authors should mention so in the manuscript.

Overall Recommendation: Major Revision

Author Response

Dear Reviewer,

Thank you for your pertinent comments. Please find attached our answers.

Kind Regards

Rémi VERNET

Round 2

Reviewer 1 Report

Authors have been responsive ! I have no further concerns 

Reviewer 2 Report

Authors have appropriately addressed the issues raised on the initial peer review. Manuscript can now be accepted for publication.

This manuscript is a resubmission of an earlier submission. The following is a list of the peer review reports and author responses from that submission.